# Anemia and associated factors among type-2 diabetes mellitus patients attending public hospitals in Harari Region, Eastern Ethiopia

**Astarekegn Bekele[1], Kedir Teji Roba[2], Gudina Egata[3], Berhe Gebremichael** [3]*

**1** Shekosh Health Center, Somali Regional State, Shekosh, Ethiopia, **2** School of Nursing and Midwifery, Haramaya University, Harar, Ethiopia, **3** School of Public Health, Haramaya University, Harar, Ethiopia

* berhegere09@gmail.com

## Abstract

### Background

Anemia is a common complication of diabetes mellitus, therefore having a major impact on the overall health and survival of diabetic patients. However, there is a paucity of evidence of anemia among diabetic patients in Ethiopia, particularly in Harari Region. Therefore, this study aimed to assess the magnitude of anemia and associated factors among Type 2 Diabetes Mellitus (T2DM) patients attending public hospitals in Harari Region, Eastern Ethiopia.

### Methods

A hospital based cross-sectional study was conducted from February 25 to March 30, 2019. Probability proportion to size sampling, followed by simple random sampling, was utilized to select 374 T2DM patients. To collect the data, mixed methods were applied using questionnaires and checklist. Participants were tested for anemia based on World Health Organization (WHO) criteria. Data was double entered to EpiData version 3.1 and exported into Stata version 14.0 for statistical analysis. Bivariate and multivariate logistic regression models were fitted; Crude Odds Ratio (COR) and Adjusted Odds Ratio (AOR) with 95% Confidence Interval (CI) were computed. Level of significance was declared at p-value less than 0.05.

### Results

The study revealed 34.8% of the participants were anemic (CI: 28.7, 40.9). Being male (AOR = 2.92, CI: 1.65, 5.17), physical inactivity (AOR = 2.58, CI: 1.50, 4.44), having nephropathy (AOR = 2.43, CI: 1.41, 4.21), poor glycemic control (AOR = 1.98, CI: 1.17, 3.34), recent history of blood loss (AOR = 4.41, CI: 1.26, 15.44) and duration of diabetes for five years and greater(AOR = 1.72, CI: 1.01, 2.96)were all significantly associated with anemia.

**Funding:** The authors received no specific funding for this work.

**Competing interests:** The authors have declared that no competing interests exist.

**Abbreviations: AOR**, Adjusted Odds Ratio; **BMI**, Body Mass Index; **CI**, Confidence Interval; **COR**, Crude Odds Ratio; **IQR**, Inter-quartile Range; **SD**, Standard Deviation; **T2DM**, Type 2 Diabetes Mellitus; **USA**, United States of America; **WHO**, World Health Organization.

## Conclusions

Anemia was a major health problem among T2DM patients in the study area. Therefore, routine screening of anemia for all T2DM patients aiding in early identification and improved management of diabetes will lead to improved quality of life in this patient population.

## Introduction

Anemia is a condition in which the number of red blood cells (and consequently their oxygen-carrying capacity) is insufficient to meet the body's physiologic needs [1]. It is a global public health problem affecting both developing and developed countries with major consequences for human health as well as social and economic development. It occurs at all stages of the life cycle [2], and affects nearly two billion (27%) people worldwide[3].

Anemia is a common complication of diabetes mellitus, and the risk of anemia in diabetic patients is estimated to be two to three times higher than that of patients without diabetes [4]. Globally, the prevalence of concurrent anemia and diabetes mellitus (both type 1 and type 2) ranges from 14% to 45% in various ethnic populations worldwide [5]. The magnitude of anemia among T2DM patients varies among studies and regions, ranging from 7.7% in the United States of America (USA) to 67%in India [6–20].

The etiology of anemia in diabetes is multifactorial and includes nutritional deficiencies, inflammation, concomitant autoimmune diseases, advanced age, lower Body Mass Index (BMI), longer duration of diabetes, peripheral vascular disease, specific medications, and hormonal changes in addition to kidney disease [21–24]. Various studies revealed the development of anemia in T2DM patients is significantly associated with sex [7, 14, 16], age [10, 14, 21], marital status [21], educational status [10], BMI, hypertension, hematological diseases [9], glycemic control, gastrointestinal disorders, and chronic kidney diseases [25]. The duration of diabetes [14] and micro-vascular complications of diabetes such as diabetic nephropathy, neuropathy and retinopathy [10, 13, 14, 17, 26, 27] have all found to be significantly associated with anemia in T2DM patients. Specific medications such as Angiotensin Converting Enzyme Inhibitors (ACEI), angiotensin receptor blockers and insulin have also been found to have a significant association with anemia in T2DM patients [13].

Serious complications of early onset anemia in diabetic patients include severe symptomatic neuropathy leading to efferent sympathetic denervation of the kidney and possible damage to the renal interstitial and inability to produce appropriate erythropoietin, systemic inflammation and inhibition of erythropoietin release [28, 29]. Anemia is found to contribute to the development and progression of micro and macro-vascular complications of diabetes, which has a negative impact on the quality of life and an additional burden on the health of the patients [7, 15, 30, 31].

Despite all these facts, anemia inT2DM remains unrecognized and untreated in 25% of the diabetic patients [8, 21]because both share similar symptoms such as lethargy, pale skin, chest pain, irritability, numbness/coldness in the hands and feet, tachycardia, shortness of breath and headache [23].

Ethiopia is one of the developing countries where both anemia and diabetes mellitus are major public health issues [32, 33]. The country is currently implementing the second National Nutrition Program (NNP-II), and is committed to addressing the major nutritional issues and improve the delivery of nutrition services for communicable and non-communicable/lifestyle related diseases by 2020 [34]. Although the patient attendance rates and medical admissions

related to diabetes in major hospitals have continued to increase, most of the clinical data are not timely and made available for decision makers in the country [21]. As a result, there is little evidence on anemia among diabetic patients in Ethiopia, particularly in the study area [21, 35]. Therefore, the aim of this study was to assess the magnitude of anemia and associated factors among T2DM patients attending public hospitals in Harari region, Eastern Ethiopia.

## Methods and materials

### Study setting and population

A hospital based cross-sectional study was conducted among T2DM patients attending public hospitals in Harari Region, Eastern Ethiopia from February 25 to March 30, 2019. Harari region is one of the nine regional states of Ethiopia, with Harar as the capital city. This region is located 526 kilometers east of Addis Ababa, the capital of Ethiopia. About 90% of the region is *Weynadega* (between 1000–1500 meters) while the rest (10%) is *Kola* (below 1000 meters). The region consists of four government (two of which are public) and two private hospitals, and five health centers [36].

The current study was conducted at the two public hospitals in the region, Jugol General Hospital and Hiwot Fana Specialized University Hospital. Jugol Hospital consists of 91 beds, with 353 administrative and 211 technical staff to provide curative, rehabilitative, and preventive health care services. Hiwot Fana Hospital, on the other hand, consists of 250 beds, with 393 technical and 248 administrative staff to also provide curative, rehabilitative, and preventive health care services. Hiwot Fana is owned and managed by Haramaya University, serving as a teaching hospital.

The study population includes T2DM patients seeking healthcare services at either Jugol or Hiwot Fana Hospital. Participants, who were seriously ill, and unable to stand and sit without assistance or support, were excluded from the study because it was difficult to get appropriate and accurate data or measurement from these patients.

### Sample size and sampling procedure

The sample size for the magnitude of anemia was determined using single population proportion formula considering the following assumptions: 95% confidence level, 5% margin of error and proportion of anemia among T2DM patients in the Sub-Saharan Africa (41.4%) [10].This provided a desired sample size of 373.

The sample size for the associated factors was calculated in Open Epi online software assuming the following assumptions: 95% confidence level, 80% power, equal unexposed to exposed ratio (1:1), proportion of anemia among patients having T2DM for five years or more (19.8%) and patients having T2DM for less than five years (9.0%) [14]. This provided a sample size of 366.

The sample size calculated for the magnitude of anemia (373) was used for this study as it was greater than the calculated sample size for associated factors. Ten percent was added for non-responders, and resulted in a final sample size of 410.

Initially, a total of 1604 patients diagnosed with T2DM fit the inclusion criteria including diagnosis of T2DM, and outpatient follow-up visit at a participating hospital (1118 in Hiwot Fana and 486 in Jugol). Sampling frames were created and the number of patients to be included in the study was determined for each hospital by probability proportion to size sampling, followed by simple random sampling (**Fig 1**).

### Data collection and measurements

Pre-tested and interviewer administered structured questionnaire was used to collect quantitative data. The questionnaire was prepared in English language and translated into local

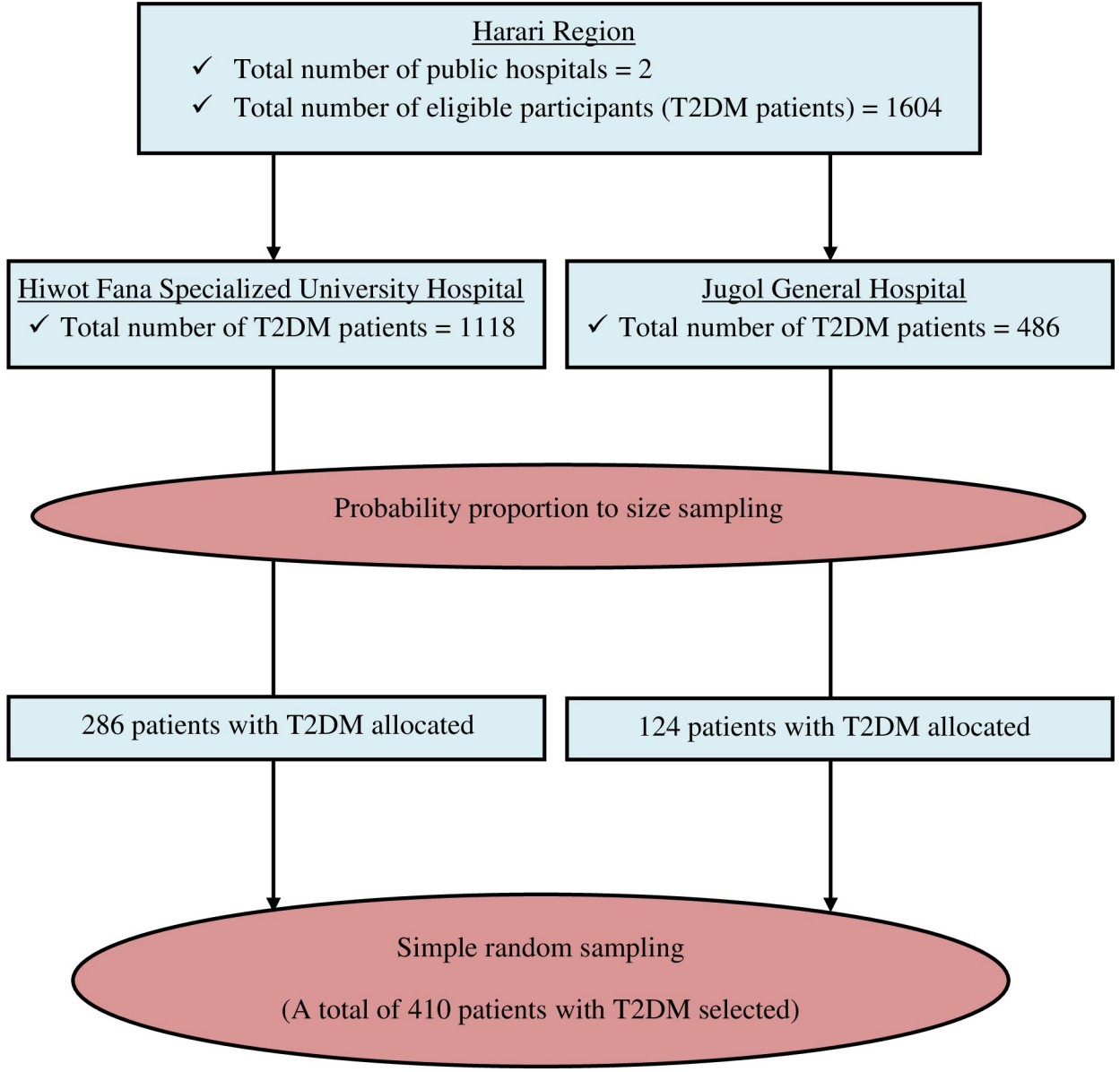

**Fig 1. Schematic presentation of sampling procedure among T2DM patients attending public hospitals in Harari Region, Eastern Ethiopia, February to March 2019.**

languages (Amharic and Afan Oromo), and back to English language. Data was collected on socio-demographic characteristics, lifestyle, household food consumption, and clinical conditions of the patients by four BSc nurses. Patient' card/chart review was also conducted to cross-check the clinical conditions of the patients, and to obtain the fasting blood sugar results done on the date of data collection.

A seven-day based food frequency questionnaire was adopted from Comprehensive Food Security and Vulnerability Analysis (CFSVA) of Ethiopia to collect data on household food consumption. A food consumption score was calculated using the frequency of consumption of various food groups by a household during the seven days preceding the interview. The seven food groups with their respective relative weights (based on relative nutritional

importance) were: cereals/grains/tubers (2), pulses (3), vegetables (1), fruits (1), meat/fish/egg (4), milk/milk products (4), sugar/honey (0.5), and fat/oil/butter (0.5). Food consumption score was calculated for the households and classified as the following: poor (0–21), borderline (21.01–34.99), and acceptable (≥35) [37].

Checklists were used to collect data from anthropometric, blood pressure, and hemoglobin measurements. Digital weight scale, which has an attached height scale (Adult Scale ASTOR), was used to measure the weight and height of the study participants. The scale was readjusted to zero and calibrated daily. During weight and height measurements, heavy clothes, shoes, bags or any other material was avoided. The height of study participants was recorded to the nearest 0.1cm. The BMI of the participants was calculated using the formula of weight (in kilogram) divided by height (in meter square), and classified as the following: underweight ($< 18.5$ kg/m$^2$), normal (18.5–24.9 kg/m$^2$), overweight (25–29.9 kg/m$^2$), and obese ($\geq$30 kg/m$^2$) [38].

Flexible and non-stretchable tape measure was used to take waist circumference measurement from the approximate midpoint between the lower margin of the last palpable rib and the top of the iliac crest, and was measured to the nearest 0.5 cm. Participants' central obesity was defined by waist circumference thresholds ($\geq$ 88 cm for women and $\geq$ 102 cm for men [39–41].

Blood Pressure was measured after the participants had rested for at least 5 minutes and taken using the left arm at the heart level, by automated digital blood pressure monitor (Omron). The mean of two measurements(at least three minutes apart) was used for analysis; hypertension was defined as blood pressure greater or equal to 140/90 mmHg or currently being on antihypertensive treatment [42, 43].

Hemoglobin levels of the participants were measured by trained laboratory technologists using HemoCue HB-301 system (HemoCue HB-301 analyzer, Sweden) and appropriate protocol. The sensitivity and specificity of the HemoCue machine was checked by complete blood count machine with five differentials (Coulter® DxH 800 analyzer). The results of hemoglobin level were adjusted to Harari Region altitude by subtracting 0.5 g/dl from all participants and for current cigarette smokers by subtracting 0.03 g/dl. Anemia was defined as a hemoglobin level less than 12 g/dl for women and 13 g/dl for men [1]

In this study, glycemic control was labeled good for the fasting blood sugar between 80 and 130 mg/dl, and poor for <80 mg/dl or ≥130 mg/dl) [44]. Likewise, participants were labeled as physically active if they participated in at least 150 minutes/week of moderate to vigorous aerobic exercise at least 3 days during the week, with no more than 2 consecutive days between bouts of aerobic activity [45].

## Data quality control

The questionnaire was translated into the local languages; i.e. Afan Oromo and Amharic for data collection and then translated back into English. Five percent of the questionnaires were pre-tested one week before the actual data collection among T2DM patients in Dilchora General Hospital, which is located about 52 Kilometers away from the study area. Data collectors were trained for two days on the data collection process to have a common understanding. Standardization of anthropometric measurements was done by considering the investigators as the gold standard. Relative Technical Error of Measurement (%TEM) was done for each data collector during the training to minimize anthropometric measurement errors. The data collectors were closely supervised by the investigators. Completeness of each questionnaire was checked by the investigators daily. Data was entered by two data clerks and consistency was cross checked by comparing the two separately entered data on EpiData.

## Data processing and analysis

After data collection, data was edited and cleaned, and each questionnaire was checked for completeness and coded. Data was entered into the computer using EpiData version 3.1 and the analysis was done using Stata version 14.0. Categorical variables were described using frequency, percentage, tables and figures. Continuous variables were assessed for normality and were described using summary measures.

Bivariable logistic regression analysis was used, and COR with 95% CI was computed to assess the association between each independent and the outcome variables. Variables with p-value <0.25 were included in the multivariable logistic regression analysis. The independent variables included in the final model were sex, educational status, alcohol use, physical activity, nephropathy, glycemic control, history of recent blood loss, and duration of diabetes. Multicollinearity test of these variables was checked using Variance Inflation Factor (VIF), and no significant (VIF > 10) collinearity was detected. Model goodness-of-fit was checked by Hosmer and Lemeshow test, and the final model was well fitted with the included independent variables (p-value = 0.89). The final model was performed to control the confounding variables and identify the associated factors, by estimating AOR with 95% CI. Statistical significance was declared at p-value < 0.05.

## Ethics approval and consent to participate

Ethical clearance was secured from Institutional Health Research Ethics Review Committee (IHRERC) of the College of Health and Medical Sciences at Haramaya University. Informed, voluntary, written and signed consent was obtained prior to initiation of the study from each participant and hospital head. The interviews and measurements were carried out privately in separate rooms. All possible identifiers were excluded from the questionnaires and checklist to ensure participants' confidentiality.

## Results

### Socio-demographic characteristics

Three hundred seventy fourT2DM patients took part in the study, resulting in a response rate of91.2%. Lost to follow-up and refusal to participate were the reasons for the non-responses. One hundred ninety eight (52.9%) participants were females. Mean age of participants was 56.3 years (Standard Deviation (SD) = 11.5), which ranges between 30–86 years. Half (50.3%) of the participants belonged to Orthodox Christianity and 135 (36.1%) are from Amhara ethnic group. Two hundred seventy (72.2%) respondents were married and 150 (40.1%) were housewives. The majority (88.5%) were living in urban areas and 101 (27%) had an education up to primary level (**Table 1**).

### Household food consumption

Cereals/grains/tubers and oil/fat/butter were the most frequently (daily) consumed food groups during the seven days preceding the interview. Accordingly, 269 (71.9%) of the participants' households consumed cereals/grains/tubers daily with a median consumption frequency of 4 times (Inter-quartile Range (IQR) = 1.0) and 190 (50.8%) consumed foods made from oil/fat/butter daily with a median consumption frequency of 4times (IQR = 2.0). However, during the seven days prior to the study, 217 (58.0%) of the households did not consume eggs, 245 (65.5%) did not consume dairy products (except butter) and 351 (93.9%) did not consume honey/sugar. Overall, 123 (32.9%), 178 (47.6%) and 73 (19.5%) of the households had poor, borderline and acceptable food consumption, respectively with a median food consumption score of 24.0 (IQR = 12.0) (**Fig 2**).

**Table 1. Socio-demographic characteristicsofT2DM patients attending public hospitals in Harari region, Eastern Ethiopia, February to March 2019 (n = 374).**

| Variables | Categories | Frequency | Percentage |
|---|---|---|---|
| Age of respondents | 30–39 years | 26 | 7.0 |
| | 40–49 years | 89 | 23.8 |
| | 50–59 years | 108 | 28.9 |
| | ≥ 60 years | 151 | 40.3 |
| Sex | Male | 176 | 47.1 |
| | Female | 198 | 52.9 |
| Ethnicity | Oromo | 114 | 30.5 |
| | Harari | 76 | 20.3 |
| | Amhara | 135 | 36.1 |
| | Tigray | 22 | 5.9 |
| | Others * | 27 | 7.2 |
| Religion | Muslim | 145 | 38.8 |
| | Orthodox | 188 | 50.2 |
| | Protestant | 41 | 11.0 |
| Marital status | Single | 28 | 7.5 |
| | Married | 270 | 72.2 |
| | Divorced | 67 | 17.9 |
| | Widowed | 9 | 2.4 |
| Occupational status | Farmer | 33 | 8.8 |
| | Housewife | 150 | 40.1 |
| | Merchant | 38 | 10.2 |
| | Governmental employer | 121 | 32.3 |
| | Others** | 32 | 8.6 |
| Educational status | Unable to read and write | 87 | 23.3 |
| | Able to read and write | 59 | 15.8 |
| | Primary level (grade 1–8) | 101 | 27.0 |
| | Secondary school (grade 9–12) | 75 | 20.0 |
| | College and above | 52 | 13.9 |
| Residence | Urban | 331 | 88.5 |
| | Rural | 43 | 11.5 |

* Wolaita, Hadiya, Somali, Gurage, Gamo

** daily laborer, Non-governmental organization worker, driver

## Lifestyle and nutritional status of the participants

The study revealed that 78 (20.9%) of the participants had a history of cigarette smoking at least once in their lifetime and 40 (10.7%) were current smokers. In addition, 108 (28.9%) of the study participants had a history of alcohol use at least once in their lifetime and 32 (8.6%) were current users. Moreover, 150 (40.1%) of the respondents were engaged in physical activity, and 119 (79.3%) of them were doing exercise three times and above per week. Regarding their nutritional status, 108 (28.9%) of the study participants were overweight and 232 (62%) had central obesity (**Table 2**).

## Clinical conditions, complications and comorbidities of T2DM

The median duration of diabetes among the participants was 5.0 years (IQR = 7.0), ranging from 1–30 years. All the patients were taking medications for the management of their

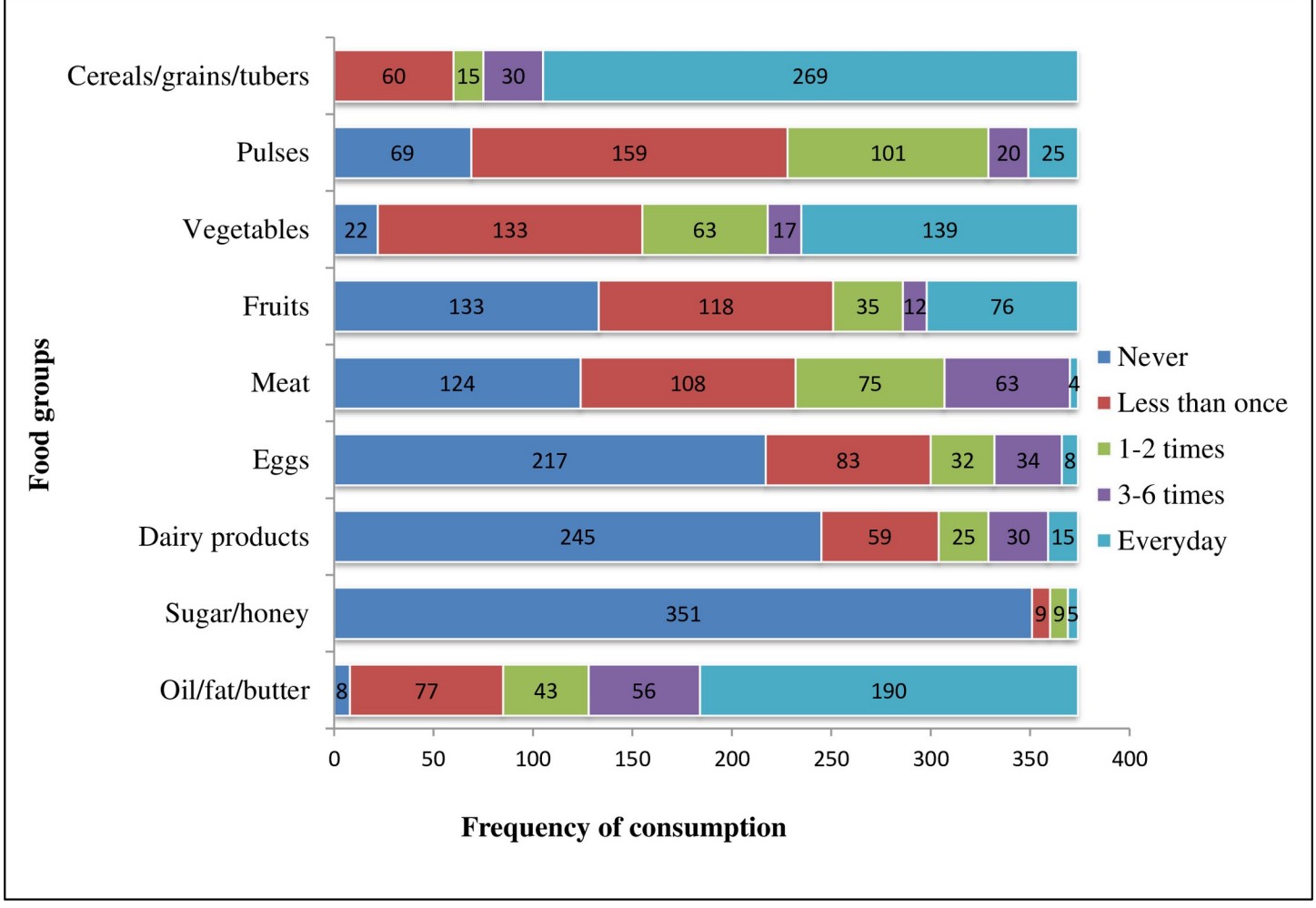

**Fig 2. Consumption frequency of different food groups (per week) among T2DM patients attending public hospitals in Harari Region, Eastern Ethiopia, February to March 2019 (n = 374).**

diabetes. Metformin was the most prescribed drug and was taken by 210 (56.1%) of the patients. In addition, 170 (45.5%) of the patients were taking medications for the management of diseases other than diabetes, and 119 (70.0%) of whom were taking anti-hypertensive drugs. Concerning the complications of diabetes, 235 (62.5%), 164 (43.9%), 169 (45.2%) and 137 (36.6%) of the patients had neuropathy, nephropathy, retinopathy and a history of hypertension, respectively. On the other hand, the glycemic control was poor in 231(61.8%) of the patients (**Table 3**).

## Magnitude of anemia and associated factors

The hemoglobin test result revealed that 130 (34.8%) of the participants were anemic (95% CI: 28.7%, 40.9%). The proportion of anemia was higher in males (41.5%) than females (28.8%). In addition, it was highest in patients aged 60 years and greater (44.4%), and lowest in those aged below 40 years (26.9%) (**Fig 3**).

Bivariate and multivariate analyses were performed in the binary logistic regression to identify factors associated with anemia. Accordingly, being male, physical inactivity, diabetic nephropathy, poor glycemic control, history of recent blood loss and duration of diabetes were

**Table 2. Lifestyle characteristicsofT2DM patients attending public hospitals in Harari region, Eastern Ethiopia, February to March2019 (n = 374).**

| Variables | Categories | Frequency | Percentage |
|---|---|---|---|
| Ever smoked cigarette | Yes | 78 | 20.9 |
| | No | 296 | 79.1 |
| Smoking currently | Yes | 40 | 10.7 |
| | No | 334 | 89.3 |
| Amount of current cigarette smoking per day (n = 40) | ¼ packet | 17 | 42.5 |
| | ½ packet | 15 | 37.5 |
| | ≥1 packet | 8 | 20.0 |
| Ever consumed alcohol | Yes | 108 | 28.9 |
| | No | 266 | 71.1 |
| Drinking alcohol currently | Yes | 32 | 8.6 |
| | No | 342 | 91.4 |
| Frequency of current alcohol drinking (per week) (n = 32) | Once | 7 | 21.9 |
| | Two times | 15 | 46.9 |
| | ≥ 3 times | 10 | 31.3 |
| Amount of drinking alcohol in typical days in bottle (n = 32) | 1–2 | 14 | 43.8 |
| | 3–4 | 10 | 31.2 |
| | ≥5 | 8 | 25.0 |
| Engaged in physical activity (n = 374) | Yes | 150 | 40.1 |
| | No | 224 | 59.9 |
| Frequency of physical exercise per week (n = 150) | Once | 8 | 5.3 |
| | Two times | 23 | 15.4 |
| | ≥ 3 times | 119 | 79.3 |
| Types of physical activity (n = 150) | Brisk walking | 142 | 94.7 |
| | Running | 51 | 34.0 |
| Nutritional status(n = 374) | Underweight | 5 | 1.3 |
| | Normal | 213 | 57.0 |
| | Overweight | 108 | 28.9 |
| | Obese | 48 | 12.8 |
| Central obesity(n = 374) | Yes | 232 | 62.0 |
| | No | 142 | 38.0 |

identified as associated factors of anemia among T2DM patients. Male patients were 2.92 times more likely to have anemia than females (AOR = 2.92, CI: 1.65, 5.17). The odds of ane- miawas2.58 times higher among patients who were not practicing the minimum recom- mended exercise (AOR = 2.58, CI: 1.50, 4.44).The occurrence of anemia was 2.43 times higher among patients who had nephropathy (AOR = 2.43, CI: 1.41, 4.21) and 1.98 times higher among patients with poor glycemic control (AOR = 1.98, CI: 1.17, 3.34). Patients with a recent history of blood loss (during three months prior to the study) were 4.41 times more likely to develop anemia (AOR = 4.41, CI: 1.26, 15.44). Furthermore, the odds of anemia was1.72 times higher among patients having diabetes for five years and above (AOR = 1.72, CI: 1.01, 2.96) (Table 4).

## Discussion

The findings of this study assessed the magnitude and associated factors of anemia among T2DM patients attending public hospitals in Harari Region, Eastern Ethiopia. Accordingly, the hemoglobin test result showed that 34.8% of the study participants were anemic. Being

**Table 3. Clinical conditions ofT2DM patients attending public hospitals in Harari region, Eastern Ethiopia, February to March2019 (n = 374).**

| Variables | Categories | Frequency | Percentage |
|---|---|---|---|
| Duration of diabetes (in years) | < 5 | 196 | 52.4 |
| | ≥ 5 | 178 | 47.6 |
| Medications taken for diabetes | Metformin | 210 | 56.1 |
| | Metformin and glibenclamide | 136 | 36.4 |
| | Insulin | 55 | 14.7 |
| Medications taken for diseases other than diabetes | Yes | 170 | 45.5 |
| | No | 204 | 54.5 |
| Types of medications taken for diseases other than diabetes (n = 170) | ACEI | 71 | 41.8 |
| | Beta blockers | 17 | 10.0 |
| | NSAIDs | 88 | 51.8 |
| | Omeprazole | 33 | 19.4 |
| | Others * | 99 | 26.5 |
| Types of diseases for which other medications taken (n = 170) | Hypertension | 119 | 70.0 |
| | Heart failure | 17 | 10.0 |
| | Gastritis | 33 | 19.4 |
| | HIV/AIDS | 13 | 7.6 |
| | Others** | 56 | 14.9 |
| Neuropathy | Yes | 235 | 62.8 |
| | No | 139 | 37.2 |
| Retinopathy | Yes | 169 | 45.2 |
| | No | 205 | 54.8 |
| Nephropathy | Yes | 164 | 43.9 |
| | No | 210 | 56.1 |
| Macro-vascular complications | Yes | 33 | 8.8 |
| | No | 341 | 91.2 |
| History of hypertension | Yes | 137 | 36.6 |
| | No | 237 | 63.4 |
| Ever tested for HIV/AIDS | Yes | 350 | 93.6 |
| | No | 24 | 6.4 |
| HIV status of the patient (n = 350) | Positive | 12 | 3.4 |
| | Negative | 338 | 96.6 |
| History of blood loss | Yes | 15 | 4.0 |
| | No | 359 | 96.0 |
| Glycemic control | Good | 143 | 38.2 |
| | Poor | 231 | 61.8 |
| Systolic blood pressure (mmHg) | <140 | 270 | 72.2 |
| | ≥140 | 104 | 27.8 |
| Diastolic blood pressure (mmHg) | <90 | 330 | 88.2 |
| | ≥90 | 44 | 11.8 |

**NSAIDs**: Non-steroidal Anti-inflammatory Drugs

* amoxicillin, ciprofloxacin, neurobion, nifidipine, hydrochlorothiazide, atorvastatin and Tenofovir/Lamivudine/Efavirenz

** peripheral neuropathy, urinary tract infections, blood loss and pneumonia

male, physical inactivity, diabetic nephropathy, poor glycemic control, history of recent blood loss and duration of diabetes were identified as associated factors of anemia among T2DM patients.

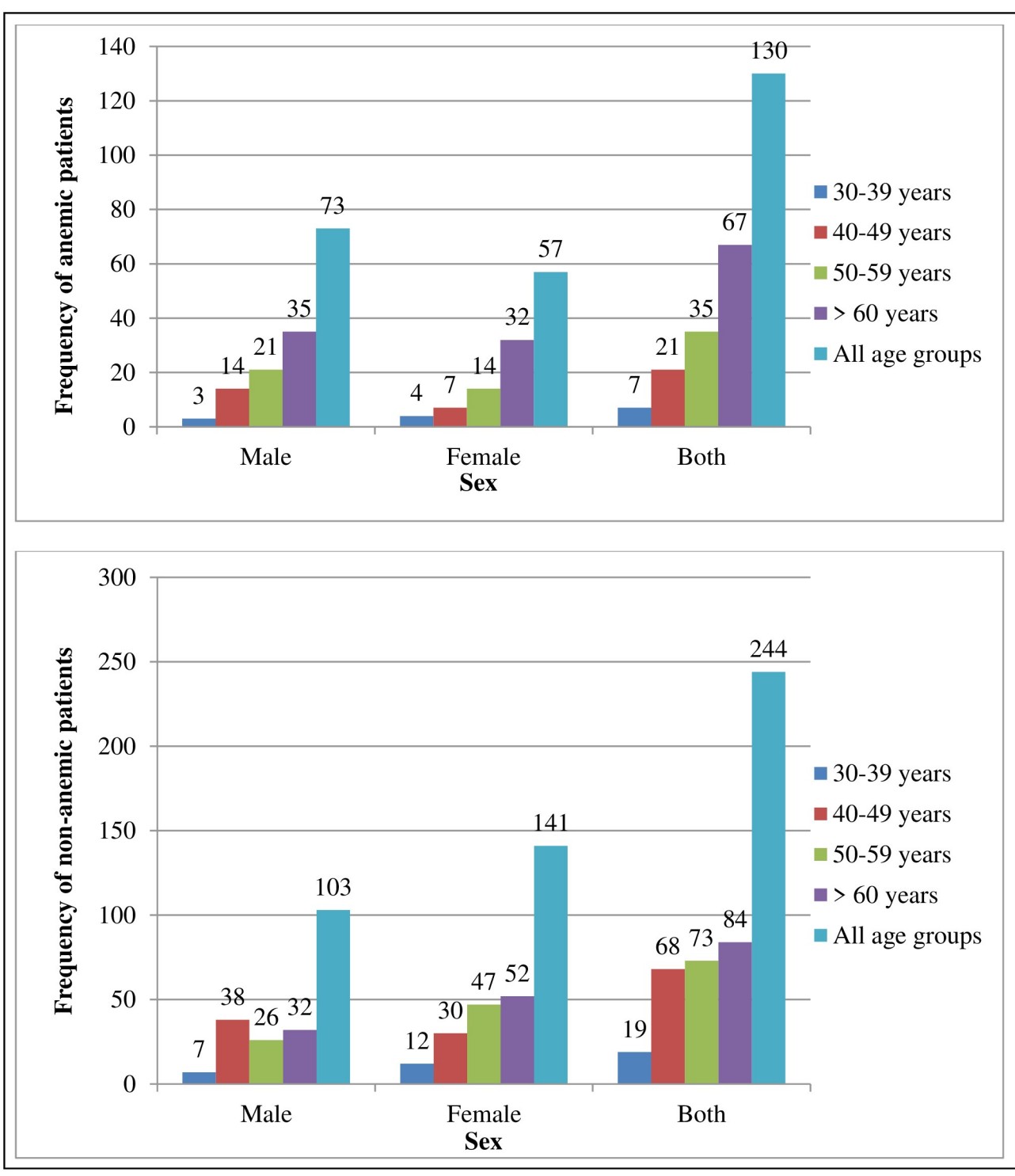

**Fig 3. Sex and age distribution among anemic and non-anemic T2DM patients attending public hospitals in Harari Region, Eastern Ethiopia, February to March 2019 (n = 374).**

The magnitude of anemia in this study was similar to reports from Iran (30.4%) [13], Malaysia (31.7% and 39.4%) [17, 18], and Brazil (34.2%) [9]. However, it was higher than other

**Table 4. Factors associated with anemia among T2DM patients attending public hospitals in Harari region, Eastern Ethiopia, February to March 2019.**

| Variables | Category | Anemia | | COR (95% CI) | AOR (95% CI) |
|---|---|---|---|---|---|
| | | Yes | No | | |
| Sex | Male | 73 (41.5%) | 103 (58.5%) | 1.75 (1.14, 2.69) * | 2.92(1.65, 5.17) ** |
| | Female | 57 (28.8%) | 141 (71.2%) | 1 | 1 |
| Educational status | No formal education | 58 (39.7%) | 88 (60.3%) | 1.80 (1.08, 3.01)* | 1.99 (0.98, 4.04) |
| | Primary level | 38 (37.6%) | 63 (62.4%) | 1.65 (0.94, 3.00) | 1.52 (0.77, 3.02) |
| | Secondary level/above | 34 (26.8%) | 93 (73.2%) | 1 | 1 |
| Ever drink alcohol | Yes | 46 (42.6%) | 62 (57.4%) | 1.61 (1.01, 2.55) * | 1.55 (0.88, 2.74) |
| | No | 84 (31.6%) | 182 (68.4%) | 1 | 1 |
| Physical activity | Yes | 28 (18.7%) | 122 (81.3%) | 1 | 1 |
| | No | 102 (45.5%) | 122 (54.5%) | 3.64 (2.24, 5.93) * | 2.58 (1.50, 4.44)* |
| Nephropathy | Yes | 80 (48.8%) | 84 (51.2%) | 3.05(1.96, 4.74)* | 2.43(1.41, 4.21)* |
| | No | 50 (16.1%) | 160 (83.9%) | 1 | 1 |
| Glycemic control | Yes | 33(22.9%) | 110(77.1%) | 1 | 1 |
| | No | 97(42%) | 134(58%) | 2.4(1.51, 3.86)* | 1.98(1.17, 3.34) * |
| History of recent blood loss | Yes | 10 (66.7%) | 5 (33.3%) | 3.98 (1.33, 11.94)* | 4.41(1.26, 15.44) * |
| | No | 120 (35.3%) | 220 (64.7%) | 1 | 1 |
| Duration of diabetes | < 5 years | 46 (23.5%) | 150 (77.5%) | 1 | 1 |
| | ≤ 5 years | 84 (47.2%) | 94 (52.8%) | 2.91(1.87, 4.54) * | 1.72 (1.01, 2.96) * |

* Statistically significant at p-value = 0.05–0.01

** statistically significant at p-value < 0.001

reports from various regions of the world, ranging from7.7% in the USA to 22% in China [7, 11, 12, 14, 19]. On the other hand, the magnitude of anemia in this study was lower compared to the findings of other studies conducted in different corners of the world, varying between 41.1% in Sub-Saharan Africa (Cameroon) and 67% in India [6, 10, 15, 16, 20]. The variations might be due to differences in socio-demographic and economic status, access to health care services, feeding habits, lifestyle, scale of study, measurements, and altitude.

The current study revealed that anemia was more likely to occur in male patients compared to females, which is in line with a study conducted in USA [7]. However, this finding was contradicting with other studies done in India and Pakistan [14, 16].Since the mean age of the participants in this study was 56 years, there may be decrement in the occurrence of anemia in females due to reduced blood loss as a result of menopause.

The odds of anemia was higher among patients who were not practicing physical activity compared to their counterparts. Physical activity increases muscle capillary density, promotes rapid enhancement of skeletal muscle oxidative capacity, insulin sensitivity, and glycemic control in adults with T2DM which are all reversible with detraining [45]. Daily exercise is recommended for patients with T2DM, to help with control of glucose and long-term consequences of the disease.

The finding of this study showed that anemia was more likely among patients having nephropathy and this is consistent with the study done in Malaysia and Iran [13, 17]. Erythropoietin is a hormone made by the kidney and if the kidney is not working properly, there may not be sufficient hormone produced [46].

The study demonstrated that patients with poor glycemic control were more likely to develop anemia compared to those with good glycemic control. This is in line with a study done in Pakistan [16]. The possible justification could be nephropathy is a well-known complication of poorly controlled diabetes. Moreover, other factors are suggested for the earlier onset

of anemia in diabetic patient including severe symptomatic neuropathy leading to efferent sympathetic denervation of the kidney along with loss of appropriate erythropoietin; systemic inflammation; and inhibition of erythropoietin release [20].

Patients with recent history of blood loss were more likely to have anemia compared to their counterparts. Likewise, the odds of anemia was higher among patients with diabetes for a duration of five years and greater which is supported by a study done in India[14]. This could be explained as many patients in Ethiopia take metformin which can interfere with cyanocobalamin absorption resulting in a metformin-induced vitamin B12 deficiency. The drug may inhibit calcium-dependent absorption of the vitamin and the intrinsic factor complex at the terminal ileum. It may also contribute to glucose-6-phosphate dehydrogenase (G6PD) mediated hemolysis [47].

Ethiopia is currently implementing a second national nutrition program, which includes addressing micronutrient deficiencies and non-communicable diseases related to malnutrition [34]. This study showed that more than one-third of the patients with T2DMhad anemia, which further complicates the diabetics overall health. Although the etiology of anemia likely is complicated and multifactorial in the diabetic patient, this study has found some specific associated factors which can be addressed to reduce the burden of anemia in this population. Several of the factors such as physical inactivity, diabetic nephropathy, and poor glycemic control are likely related to poor management of their diabetes and subsequent progression of damage as a result of unregulated glucose levels. This is the opportune time to address better control of T2DM through diet, and exercise in addition to pharmacological treatment.

## Strength and limitations of the study

The strength of the study is that it was laboratory based using the measurement of hemoglobin. However, since it is a cross-sectional study, it will not show the temporal relation between the independent and dependent variables. There is also a possibility of recall bias since the questionnaire for food consumption was based on recall knowledge.

## Conclusions

More than one-third of the T2DM patients included in this study had anemia. The burden of anemia was higher in male and elderly patients. Being male, physical inactivity, diabetic nephropathy, poor glycemic control, history of recent blood loss and duration of diabetes were identified as associated factors of anemia among T2DM patients. Therefore, there should be routine screening of anemia for all diabetic patients in health institutions or diabetic clinics for early diagnose and to manage anemia, thereby improving patient's quality of life. Interventions such as physical activity (aerobic exercise), early diagnosis and treatment of comorbidities and complications, and appropriate glycemic control can also benefit diabetic patients in reducing anemia.

## Implications of the study

This study was hospital based and cannot be generalized to the community. Additionally, it may not influence national level nutrition policy or program since it was small scale study. However, it can be utilized as an input, with other similar studies, to produce pooled national or international estimates for policy decision making. Besides, it can be useful as baseline information for further epidemiological and nutritional studies in similar settings. Furthermore, the results of this study can help clinicians in decision making regarding anemia diagnosis among T2DM patients in the study area or similar settings.

## Supporting information

**S1 Dataset. The dataset from which the results of the study were produced (SPSS file).**
(SAV)

**S2 Dataset. The dataset from which the results of the study were produced (Stata file).**
(DTA)

**S1 Questionnaire. The data collection tool (questionnaire and checklist) in English.**
(DOCX)

**S2 Questionnaire. The data collection tool (questionnaire and checklist) in Amharic language.**
(DOCX)

**S3 Questionnaire. The data collection tool (questionnaire and checklist) in Afan Oromo.**
(DOCX)

## Acknowledgments

The authors are grateful to the staff of Jugol General Hospital and Hiwot Fana Specialized University Hospital, the data collectors, supervisors, study participants, and questionnaire translators for their cooperation. The authors are also grateful to Dr. Tara D'Ann Wilfong for her contribution in revising the manuscript particularly language copyediting.

## Author Contributions

**Conceptualization:** Astarekegn Bekele, Kedir Teji Roba, Gudina Egata, Berhe Gebremichael.

**Data curation:** Astarekegn Bekele, Kedir Teji Roba, Gudina Egata, Berhe Gebremichael.

**Formal analysis:** Astarekegn Bekele, Kedir Teji Roba, Gudina Egata, Berhe Gebremichael.

**Investigation:** Astarekegn Bekele.

**Methodology:** Astarekegn Bekele, Kedir Teji Roba, Gudina Egata, Berhe Gebremichael.

**Project administration:** Astarekegn Bekele.

**Resources:** Astarekegn Bekele.

**Software:** Astarekegn Bekele, Kedir Teji Roba, Gudina Egata, Berhe Gebremichael.

**Supervision:** Astarekegn Bekele, Kedir Teji Roba, Gudina Egata, Berhe Gebremichael.

**Validation:** Astarekegn Bekele.

**Writing – original draft:** Astarekegn Bekele.

**Writing – review & editing:** Astarekegn Bekele, Kedir Teji Roba, Gudina Egata, Berhe Gebremichael.

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
