## [Decision Letter · Decision Letter 0]

14 Oct 2019

PONE-D-19-24012

Anemia and associated factors among type-2 diabetes mellitus patients attending public hospitals in Harari Region, Eastern Ethiopia

PLOS ONE

Dear Mr Gebremichael,

Thank you for submitting your manuscript to PLOS ONE. After careful consideration, we feel that it has merit but does not fully meet PLOS ONE’s publication criteria as it currently stands. Therefore, we invite you to submit a revised version of the manuscript that addresses the points raised during the review process.

We would appreciate receiving your revised manuscript by Nov 28 2019 11:59PM. To enhance the reproducibility of your results, we recommend that if applicable you deposit your laboratory protocols in protocols.io, where a protocol can be assigned its own identifier (DOI) such that it can be cited independently in the future. For instructions see: http://journals.plos.org/plosone/s/submission-guidelines#loc-laboratory-protocols

We look forward to receiving your revised manuscript.

Kind regards,

Naeti Suksomboon

Academic Editor

PLOS ONE

**Journal Requirements:**

2. Our editorial staff has assessed your submission, and we have concerns about the grammar, usage, and overall readability of the manuscript.  We therefore request that you revise the text to fix the grammatical errors and improve the overall readability of the text before we send it for review. We suggest you have a fluent, preferably native, English-language speaker thoroughly copyedit your manuscript for language usage, spelling, and grammar.

If you do not know anyone who can do this, you may wish to consider employing a professional scientific editing service.  

Whilst you may use any professional scientific editing service of your choice, PLOS has partnered with both American Journal Experts (AJE) and Editage to provide discounted services to PLOS authors. Both organizations have experience helping authors meet PLOS guidelines and can provide language editing, translation, manuscript formatting, and figure formatting to ensure your manuscript meets our submission guidelines. To take advantage of our partnership with AJE, visit the AJE website (http://learn.aje.com/plos/) and enter referral code PLOS15 for a 15% discount off AJE services. To take advantage of our partnership with Editage, visit the Editage website (www.editage.com) and enter referral code PLOSEDIT for a 15% discount off Editage services. If the PLOS editorial team finds any language issues in text that either AJE or Editage has edited, the service provider will re-edit the text for free.

Please note that PLOS ONE does not copyedit accepted manuscripts and that one of our criteria for publication is that articles must be presented in an intelligible fashion and written in clear, correct, and unambiguous English (http://www.plosone.org/static/publication#language). If the language is not sufficiently improved, we may have no choice but to reject the manuscript without review.

3. Please provide further details concerning the pretesting of the questionnaire involved in this study, i.e. who was this tested on and on how many?

**Comments to the Author**

1. Is the manuscript technically sound, and do the data support the conclusions?

Reviewer #1: Yes

Reviewer #2: Partly

2. Has the statistical analysis been performed appropriately and rigorously? 

Reviewer #1: Yes

Reviewer #2: Yes

3. Have the authors made all data underlying the findings in their manuscript fully available?

Reviewer #1: Yes

Reviewer #2: Yes

4. Is the manuscript presented in an intelligible fashion and written in standard English?

Reviewer #1: Yes

Reviewer #2: No

5. Review Comments to the Author

Reviewer #1: Bekele et al. aimed to assess the magnitude of anemia and associated factors among Type 2 Diabetic patient. They conduct a cross-sectional study to collect 374 T2D patients attending public hospitals in Harari Region, Eastern Ethiopia. This study concluded that anemia was a major health problem and routine screening of anemia for T2D patients were recommended. The manuscript was well prepared. I have only some minor comments.

1. Some exclusions have been applied in the sample recruitment. Please provide the reasoning for that.

2. In the sampling procedure section, the sample size of 410 is planned. But at the end, only 374 (line 209) were analyzed. Please clarify the discrepancy.

3. Line 260. The wording “the magnitude of anemia… was 130” reads awkward.

Reviewer #2: This study is notable for its use of in person questionnaires in a sampled population to determine the risk factors associated with anemia among hospitalized patients with T2DM. Interviews were very in depth including a seven day food frequency questionnaire. This data is interesting in the context of the nutrition programs currently underway in Ethiopia.

The authors state that that anemia is a major problem in Ethiopia. It would build the rationale for this study if they are able to reference any data supporting this in the introduction.

The methods require clarification in writing. The authors state that the study population was comprised of all T2DM patients, but they also applied a sampling scheme to select the analysis population. Overall, the data appears to be analyzed appropriately to reach the reported conclusions. For adjusted odds ratios, the authors should be sure to report what these estimates have been adjusted for (i.e. what was included in the multivariate model)

What data is available regarding the admission reason for the study participants? Are these results generalizable to individuals who have T2DM but were not hospitalized?

In the discussion, the authors report both that physical activity is linked to anemia (line 307), which does not match their results table, and that physical inactivity is linked to anemia (line 337). Please clarify.

Editing is needed particularly with respect to lines: 45 (“having faulty in the blood cells”), 73 (clarify in which cases similar signs and symptoms are observed) and in a number of places in the discussion.

6. PLOS authors have the option to publish the peer review history of their article (what does this mean?). If published, this will include your full peer review and any attached files.

Reviewer #1: No

Reviewer #2: No

---

## [Author Response · Author response to Decision Letter 0]

31 Oct 2019

General information: For the comments and questions raised by the academic editor and reviewers, we have uploaded point-by-point response letter, manuscript with track changes, and clean manuscript. In addition, we uploaded a separarte language edited manuscript with track changes as supporting information.

1. Comments/questions of academic editor and responses

Comment-1: Please ensure that your manuscript meets PLOS ONE's style requirements, including those for file naming.

Response: The revised manuscript has already been modified to PLOS ONE's style

Comment-2: Our editorial staff has assessed your submission, and we have concerns about the grammar, usage, and overall readability of the manuscript. We therefore request that you revise the text to fix the grammatical errors and improve the overall readability of the text before we send it for review. We suggest you have a fluent, preferably native, English-language speaker thoroughly copyedit your manuscript for language usage, spelling, and grammar.

Response: The revised manuscript has been copyedited by Dr. Tara D’Ann Wilfong. She is a native English language speaker and is from University of Florida. She is global public health specialist with a medical degree background. The language copyedited manuscript with track changes has been uploaded as supporting information

Comment-3: Please provide further details concerning the pretesting of the questionnaire involved in this study, i.e. who was this tested on and on how many?

Response: It was conducted on 5% of the questionnaires in DilChora General Hospital which is located in Dire Dawa city, eastern Ethiopia. We incorporated it in the revised manuscript on page 9, lines 178-180

Comment-4: Please include captions for your Supporting Information files at the end of your manuscript, and update any in-text citations to match accordingly.

Response: Corrected accordingly

2. Comments/questions of reviewer #1 and responses

Comment-1: . Some exclusions have been applied in the sample recruitment. Please provide the reasoning for that.

Response: The reason is it was difficult to interview the seriously ill participants and was difficult to take anthropometric data from individuals who can not sit or stand without support. We included this in the revised manuscript on page 5, line 106-107

Comment-2: In the sampling procedure section, the sample size of 410 is planned. But at the end, only 374 (line 209) were analyzed. Please clarify the discrepancy.

Response: It was due to lost to follow up from their visit/appointment and refusal not to participate. To compensate this non-responders, we added 10% to the final sample size before starting the data collection. We have included this on page 10, line 215-216.

Comment-3: Line 260. The wording “the magnitude of anemia… was 130” reads awkward

Response: We have modified it and is on page 17, line 266 in the revised manuscript

3. Comments/questions of reviewer #2

Comment-1: The authors state that that anemia is a major problem in Ethiopia. It would build the rationale for this study if they are able to reference any data supporting this in the introduction.

Response: We have put citations for this information on page 4, line 79

Comment-2: The methods require clarification in writing. The authors state that the study population was comprised of all T2DM patients, but they also applied a sampling scheme to select the analysis population. Overall, the data appears to be analyzed appropriately to reach the reported conclusions. For adjusted odds ratios, the authors should be sure to report what these estimates have been adjusted for (i.e. what was included in the multivariate model)

Response: We have modified the method section in the revised manuscript, particularly the study population and data analysis sections

Comment-3: What data is available regarding the admission reason for the study participants? Are these results generalizable to individuals who have T2DM but were not hospitalized?

Response: The hospitals have diabetic clinics where diabetic patients are followed as outpatients. So, we directily collected the data from the patients in the diabetic clinics while the patients come for their follow-up. However, the results of this study could not be generalized for diabetics at community level because the study was conducted at hospital level..

Comment-4: In the discussion, the authors report both that physical activity is linked to anemia (line 307), which does not match their results table, and that physical inactivity is linked to anemia (line 337). Please clarify.

Response: Thank you for your observation; that is a editorial error. We have now made it consistent in the revised manuscript

Comment-5: Editing is needed, particularly with respect to lines: 45 (“having faulty in the blood cells”), 73 (clarify in which cases similar signs and symptoms are observed) and in a number of places in the discussion.

Response: We have modified lines 45 and 73. In the revised manuscript, the modifications are on lines (44-45) and (75-77), respectively. For the language errors, we have made the manuscript be reviewed and edited by native English speaker. We have also corrected other technical errors.

---

## [Decision Letter · Decision Letter 1]

12 Nov 2019

Anemia and associated factors among type-2 diabetes mellitus patients attending public hospitals in Harari Region, Eastern Ethiopia

PONE-D-19-24012R1

Dear Dr. Gebremichael,

We are pleased to inform you that your manuscript has been judged scientifically suitable for publication and will be formally accepted for publication once it complies with all outstanding technical requirements.

With kind regards,

Naeti Suksomboon

Academic Editor

PLOS ONE

Additional Editor Comments (optional):

Reviewers' comments:

Reviewer's Responses to Questions

**Comments to the Author**

1. If the authors have adequately addressed your comments raised in a previous round of review and you feel that this manuscript is now acceptable for publication, you may indicate that here to bypass the “Comments to the Author” section, enter your conflict of interest statement in the “Confidential to Editor” section, and submit your "Accept" recommendation.

Reviewer #1: All comments have been addressed

2. Is the manuscript technically sound, and do the data support the conclusions?

Reviewer #1: (No Response)

3. Has the statistical analysis been performed appropriately and rigorously? 

Reviewer #1: (No Response)

4. Have the authors made all data underlying the findings in their manuscript fully available?

Reviewer #1: (No Response)

5. Is the manuscript presented in an intelligible fashion and written in standard English?

Reviewer #1: (No Response)

6. Review Comments to the Author

Reviewer #1: (No Response)

7. PLOS authors have the option to publish the peer review history of their article (what does this mean?). If published, this will include your full peer review and any attached files.

Reviewer #1: No

---

## [Editor Report · Acceptance letter]

25 Nov 2019

PONE-D-19-24012R1 

Anemia and associated factors among type-2 diabetes mellitus patients attending public hospitals in Harari Region, Eastern Ethiopia 

Dear Dr. Gebremichael:

I am pleased to inform you that your manuscript has been deemed suitable for publication in PLOS ONE. Congratulations! Your manuscript is now with our production department. 

With kind regards,

on behalf of

Dr. Naeti Suksomboon 

Academic Editor

PLOS ONE